# Five Different Lives after Suffering from Spinal Cord Injury: The Experiences of Nurses Who Take Care of Spinal Cord Injury Patients

**DOI:** 10.3390/ijerph19031058

**Published:** 2022-01-18

**Authors:** Shou-Yu Wang, Shih-Ru Hong, Jung-Ying Tan

**Affiliations:** 1Discipline of Nursing, School of Health, University of New England, Armidale 2351, Australia; cindy.wang@une.edu.au; 2Department of Nursing, Farlin Hospital, Changhua 508, Taiwan; 3College of Nursing, Hungkuang University, Taichung 433, Taiwan; jytan@sunrise.hk.edu.tw

**Keywords:** experience of spinal cord injury, nurses, qualitative research, phenomenology

## Abstract

According to statistics of Ministry of Health and Welfare, accidents were the sixth common causes of death in Taiwan in 2016. A total of 1200 new cases of spinal cord injury each year are caused by accidents and adverse effects. This study explored nurses’ experience of caring for patients with spinal cord injury. Hermeneutic phenomenology was used in the data analysis. The five themes emerged: dramatic changes in life, life lost control, life after catastrophic injury, life takes turns for the worse for family, and the power of rebirth. This study revealed that accidents were the primary cause of spinal cord injury, and that nurses may neglect patients’ mental and social care. Career guidance should be offered to spinal cord injury patients to ensure that they retain sociality. This study provides recommendations regarding a patient’s and their family’s post-injury adaption process. The sexual desire of patients should be further explored in future research.

## 1. Introduction

According to the 2016 World Health Organization (WHO) statistics, approximately 2.5–5 million cases of spinal cord injury occur annually worldwide [1]. The Republic of China Spinal Cord Injury (SCI) Patient Alliance estimated that approximately 1000–1200 spinal cord injuries occur every year in Taiwan [2]. Apart from providing medical assistance, strengthening patients’ psychological healthcare; providing nursing plans, execution, and education; and considering suitable medical services for patients and their caregivers are the other important nursing goals [3,4]. These goals will assist patients to re-recognize their disabilities and return to their families and communities as soon as possible to face the future positively and have a better quality of life [5].

Therefore, our study aimed to help clinical nurses to identify difficulties of care during the care process and to provide a reference for future clinical education for nurses to improve the care competency of nurses and care quality of SCI patients.

## 2. Background

Nursing professionals are a part of multidiscipline team, and the provision of proper nursing care can prevent or alleviate further SCI and promote the best outcomes for patients [6]. The period of a psychological analysis of nursing professionals who care for patients with SCI conducted in the UK reported that these nurses are exposed to a highly demanding and stressful clinical environment. The study found that these nurses require psychological support, stress sharing, and confidence building [7]. With regards to nursing, it was reported that more care for patients with SCI and their family members is needed, and that further health education should be provided when necessary. It was also found that other than in clinical care, there was very little evidence of nurses on patient-related issues when caring for the psychological aspect of patients or in patient care experiences.

From physiological perspectives, studies from Taiwan and China have compared the urinary system and quality of life of SCI patients. Studies have found that 39.7% of patients have no spontaneous or reflex urinary function and require urinary catheters, 22.9% opted for long-term indwelling urinary catheter and bladder fistula, and 36.11% are dependent on urinary catheters for urination. The incidence of urinary tract infection due to long-term indwelling urinary catheter was 43.9% [8,9]. As the severity of injury varies among patients with spinal cord injury, these patients suffer from long-term mobility problems, resulting in defects in urination control and increases in the incidence of infections due to long-term indwelling urinary catheters [8]. Therefore, pressure injury prevention and urinary tract infection management are important.

In studies concerning psychology and quality of life, the World Health Organization (WHO) defined quality of life as an individual’s perception of their position in life in the context of the culture and value systems in which they live and in relation to their goals, expectations, standards, and concerns. Regardless of early stage or long-term injury, patients often experience complications. When complications and physical disability occur, the quality of life of patients with SCI will become an important goal [1]. Parashar (2015) [10] employed qualitative phenomenological approach to explore the meaning of hope in 20 patients with SCI, and the following three themes emerge: hope for complete recovery, hope for self-reliance despite injury, and hope for an optimum quality of life. Zhuang, Yang, and Ji (2011) [11] employed qualitative phenomenological methods to study the feelings of 10 patients with SCI regarding their injury experiences, and the following four themes were found: “shock”, “unpreparedness”, “how to live their future lives”, “lapsing into an abyss of despair”, and “reflection on the meaning of existence”.

Primary caregivers play important roles in caring for patients with SCI. When there are changes to the family structure, the primary caregivers will experience unease when faced with the health status of family members, resulting in role changes and limitations in daily life. They tend to experience physiological and psychological burden [12]. After most patients are discharged from the hospital, they often realize that their spouse has taken over the role of the primary caregiver [13]. Combination of the spouse and primary caregiver roles results in a long-term stressful state for the care-giving individual, because they need to strengthen the patient’s psychological support and care for their life. This also increases the psychological burden of primary caregivers because they are unable to obtain proper rest due to sudden care work, making them susceptible to depression [14]. In addition, Charlifue, Botticello, Kolakowsky-Hayner, Richards, and Tulsky (2016) [15] conducted qualitative interviews on the experiences of 16 primary caregivers and obtained positive and negative responses. Regarding positive aspects, the caregivers appreciated self-changes and strengthening of family cohesiveness. Physical and emotional tension, dissatisfaction with family relationships, and health-related problems (such as fatigue and lack of sleep) were the negative aspects. In addition to decreases in autonomy due to impairment of physical function in patients with spinal cord injury, the multiple effects of interpersonal relationships, social interactions, and employment simultaneously presented life challenges to primary caregivers.

During the clinical and literature review processes, we observed a lack of in-depth studies examining the experience sharing of nursing staff for patients with SCI. Therefore, this study aimed to explore the experiences of caring for patients with SCI by nursing professionals.

## 3. Purpose of the Study

This study aimed to explore the experiences of caring for patients with spinal cord injury by nursing professionals.

## 4. Research Design

In this study, we employed Heidegger’s (1962) hermeneutic phenomenology. The main purpose of using this technique was to explore the phenomena of the social world. Human experiences provide descriptions of meaning and experiences. Our study aimed to develop an in-depth understanding of the meaning, or theme of experiences, and to explore the aspects via which the meaning of the study could be understood, which is the aim of interpretative phenomenology [16,17].

### 4.1. Study Method and Procedure

Purposive sampling was employed in this study. Nurses in charge of caring for patients with SCI in a regional teaching hospital in Taichung City were recruited as participants. After literature review and joint discussions between the researchers, the interview outline was formed and one-to-one in-depth interviews were conducted. The researchers invited the interested nursing professionals to participate in the study; snowball sampling was used to identify the recommended nursing professionals. The second author (Ms. Hong) did all interviews during data collection. All authors participate in data analysis process. The authors met regularly to discuss data analysis to avoid bias in the interpretation of data. By meeting regularly during data analysis phase (data collection and data analysis happened concurrently in the study process), it is able to ensure to minimize the interviewer’s objectivity during the interviews.

The aim of the study was explained to the nurses, and study interviews were conducted. Consent form was obtained from participants before the interview was conducted. The inclusion criteria were as follows: (a). Currently employed nurses with ≥6 months of clinical experience; (b). able to speak Chinese and Taiwanese; (c). previously cared for quadriplegic or hemiplegic patients; and (d). agreed to participate recorded interviews. The exclusion criteria were as follows: (a). Nurses with <6 months of clinical experience; (b). nurses who did not consent to interview recording; and (c). nurses who did not previously care for quadriplegic or hemiplegic patients.

### 4.2. Data Analysis

People have interpretations to their life in relation to their life experiences [18]; therefore, during the interviews, open attitudes were employed, inductive and descriptive methods were used, and overall experiences were presented with no presumptions or expectations [19]. Researchers should eliminate bias and hold onto the method of epoché, understand the concept of knowledge acquisition, and explain the data collection process to the involved nursing professionals [20]. In this study, the seven steps in phenomenology research proposed by Colaizzi (1978) [21] were used, and the interview recordings were repeatedly played to generate verbatim transcripts. During the data collection process, the interview data of the first five participants were analyzed; further, similar themes of the verbatim transcripts were analyzed and categorized. During the data analysis process, the meaning codes were labeled with different colors, and the key points of every nursing professional was extracted to a reflective journal. After repeated in-depth analysis, meanings were generated, which were converted to meaningful themes. The data saturation was reached. The results were faithfully presented in this study.

The rigor of this study is based on the four items of credibility, transferability, dependability, and confirmability proposed by Lincoln and Guba (1985) [22]. These four items are the criteria used to describe the objectivity of data analysis in this study. Regarding credibility, the researchers used the verbatim transcript analysis results to present the experiences of the interviewees, establish a relationship with them, and describe their personal experiences during the interview. During the data analysis process, the researchers jointly review the accuracy of the data. Regarding transferability, the data obtained by the researchers are derived from the descriptions of different interviewees. After the researchers describe it in detail alone, the interviewees’ experiences can be transferred and used.

In addition, during the study analysis process, intact written data were generated, including literature review, interview outline, recordings, and verbatim transcripts to clearly present the researchers’ thought process and achieve dependability. In addition, regarding confirmability, repeated listening and thinking about the participants’ viewpoints were performed. All texts and voice recordings were consolidated and stored with the interviewee materials for confirmation in the future.

### 4.3. Ethical Considerations

This study was approved by the Institutional Review Board (Review no.: 105048). During the data analysis process, participants were replaced by the English letters A–O. During the data collection process, no participants withdrew from the study. No physical and psychological harms to the participants were noted. The collected verbatim transcript and analysis data were stored and encrypted as computer files in a personal computer to protect the participants’ confidentiality.

## 5. Results

In this study, we explored the experience of nursing professionals in caring for patients with SCI. The researchers performed data collection and analysis from the verbatim transcripts. The seven steps in the phenomenology research method proposed by Colaizzi (1978) [22] were used to obtain five themes, which are the five different lives of patients after suffering from SCI: “dramatic changes in life”, “life lost control”, “life after injury”, “life takes turns for the worse for family”, and “the power of rebirth”. The following section will describe the demographic information of the participants and the five themes. 

### 5.1. Demographic Information

This study was conducted from January to September 2017; total 15 nurses (age, 23–40 (mean age, 33.6) years) were interviewed. The participants included 14 females and 1 male (during data collection phase, the researchers did attempt to recruit male nurses. However, some male nurses either canceled their interviews or were not able to do the interviews due to military service); of those, 12 were married and 3 were single. The mean work experience years were 11.7. Each interview lasted for approximately 22–70 min. Results are shown in Table 1.

### 5.2. Dramatic Changes in Life

This study found that in the experiences of caring for patients with SCI who are paralyzed, most patients encountered by nurses are paralyzed due to accidents, such as Nurse D stated:

“*When I was a new nurse, my deepest impression was of a young man who is the main breadwinner of his family. He was cycling and then suddenly flew out and injured his cervical spine and became paralyzed.*” 

Nurse K mentioned in her care experience as follows:

“*A female patient underwent surgery for herniated lumbar intervertebral disc. The cause of paraplegia is incomplete spinal cord injury. In her old photographs, she wore a cheongsam. It has been 7 years and she still has not got over it.*”

From this study, anybody may encounter accidents in the course of their life. It becomes difficult for individuals to accept the life after injury that have forced them to stop in their life journey.

In this study, the experience of caring for patients with spinal cord paralysis by nurses showed that they may encounter the patient’s death or need for continuous care in the patients’ future lives, regardless of the patients being in the acute or chronic phase. Nurses face how to assist patients to overcome paralysis and suffering. In the acute phase, patients face possible paralysis. How to assist patients with face paralysis as well as how assist them in recovering maximum function in the limbs is a major challenge faced by nursing professionals.

### 5.3. Life Lost Control 

Regardless of whether the patient has quadriplegia or hemiplegia, loss of self-control is an unbearable psychological burden. Nurse A described the experience of how to assist patients to overcome psychological impact when they felt that their limbs were unable to move as usual:

“*After surgery, we returned to the intensive care unit. After wake up (from the surgery), I found that he (she) had no sensation in the lower half of the body. When the patient could not be taken off the ventilator even with respiratory training, it caused him (her) to become even more depressed, and the patient committed suicide by biting on the (endotracheal) tube.*”

The experience mentioned by Nurse B is as follows:

“*When care was provided at the acute phase, his (her) progress during spinal cord injury was slow. I continuously gave him (her) support. When I felt that he (she) is not moving, I kept encouraging him (her).*”

After SCI, care often requires encouragement to provide psychological support for patients. Nurse L mentioned:

“*At the start, everybody wants to save his (her) life. After he (she) has been saved, muscle power in the four limbs does not remain that good. We spent 2 h helping him (her) move his (her) hands and feet.*”

In the intensive care unit, due to emergency, resuscitation is the focus; further, there is less consideration for patient outcomes and subsequent physical disability. In the acute phase, a range of motion activities are conducted once every two hours to prevent limb contracture in patients with SCI.

In contrast, Nurse C observed the following from her experience:

“*I was previously placed in the intensive care unit where many patients were unable to accept the fact that they were quadriplegic or hemiplegic. These patients were intubated with endotracheal tubes and relied on the phonetic board for communication, which took a lot of time. I had to not only flip them over but also spend time communicating. This is because the patients do not know what is happening. I should at least let the patient know what has happened to them and the doctor’s treatment plan for them.*”

From the nurse’s experiences, we are able to see the impact on care in addition to maintaining professionalism in patient care. Even if they are busy with clinical work, nurses still assist patients in moving their limbs during treatment. As patients lose their original physical autonomy, time needs to be spent during nursing care to communicate with patients and assist them to understand their current status.

### 5.4. Life after Catastrophic Injury 

Due to congenital or acquired causes, regardless of whether quadriplegia caused by SCI occurs, patients are unable to recover past functions of their limbs after acute care. Only rehabilitation can maintain maximum muscle strength after such injuries. Nurse F who works in the rehabilitation ward described how patients rehabilitate and the subsequent muscle strength development during rehabilitation as follows:

“*How do patients rehabilitate? I see that they grab things and train grip strength. For legs, I sometimes see that they kick a ball or roll logs. We only hope that their joint muscles do not contract.*”

Under interactions between different specialties, in addition to assisting in rehabilitation, nurses should also know how to maintain muscle strength in patients. Under such exchanges, they will also learn that the true purpose of occupational therapy is to strengthen the remaining functions of the limbs and to use rehabilitation training to maintain muscle strength and avoid limb atrophy.

Nurse D mentioned self-learning methods, such as meditation, are used to guide patients:

“*At the start, the entire body is relaxed. You channel the entire body’s strength to the right leg and slowly meditate such that the strength is then channeled to the right leg. Unexpectedly, this was successful on one day. Finally, the patient could walk by themselves using a walker.*”

In the spirit of employing a method with a self-learning attitude that does not threaten patient safety, a psychological progressive effect on the physiology method was applied on the patients, which successfully enabled patients to gradually recover muscle strength and simultaneously increase achievement in the nursing perspective.

### 5.5. On the Other Hand, Nurse Practitioner I Mentioned

“*Spinal cord injury requires multiple psychological aspects and levels of rehabilitation during rehabilitation. This also applies to family members. Actually, both family and sexual rehabilitation are also important. However, Taiwan is not so advanced and liberal, and only few people discuss such problems.*”

In addition to maintenance of limb function in patients, it was also mentioned that due to cultural differences, very few people were concerned that considerable amount of attention should be paid to a patient’s sexual needs. In this aspect, except Nurse practitioner I, no other participants mentioned the sexual needs of patients or their views on sex after paralysis in this study.

Regardless of the treatment during the acute to chronic phase, the adaptability of patients toward the disease should be assessed at any time. After they are injured, the patients are unable to exert force on both their hands and are also unable to stand on their feet in daily life. This causes family members to experience changes as well, causing variations in their roles between family members. Many difficulties impact the lives of patients with SCI. Prompt changes perhaps cannot occur in a short period of time. Accompanying patients and caring for them during the nursing process, examining how to solve and understand factors affecting patients with SCI and family caregiver stress, and searching for resources to overcome these difficulties should be conducted.

### 5.6. Life Takes Turns for the Worse for Family

In addition to rehabilitation treatment, the roles of patients in their families will also be changed. In the phase of sudden difficulties, the key to overcome low points is to boldly face these difficulties. Family support and care becomes the strength supporting patients. An example is Nurse N who mentioned the patient’s thoughts on his/her family and self after paralysis:

“*He is a fruit farmer who accidentally fell from a tree and became paralyzed. He is optimistic because his family members are optimistic. He told his wife that there is no need to save him as their children are unmarried and this will consume their family’s finances. He is also concerned whether his children’s future partners will be able to accept their family.*”

No one can imagine themselves becoming dependent on others in their daily life when they used to be independent, and negative psychological reactions often occur. They worry that their family members will have to bear different life changes because of them.

An example was mentioned by Nurse G where things that family members would have to bear after the patient became paralyzed were stated:

“*Rehabilitation requires us to carry the patient up and down, and family members will have to follow the nurse to learn about all this. If there are sudden events, we will tell him (her) to hire a caretaker so that they can rest for a few days. Some people have financial problems and are unable to hire a caretaker. In this case, social workers will take care of this.*”

Nurse I mentioned the following during the interview:

“*Most family members are unable to accept the condition. After some time, anger may occur. This can be so because rehabilitation may not be as optimistic as they have imagined or their understanding of the disease may be insufficient.*”

During emergencies, most family members do not try to clearly understand the patient’s disease. After onset, as time passes, when they see no improvement in the recovery status of the patient’s limbs, they will feel angry. Care is not only limited to the professional staff. Assistance from family members is required even in the clinical setting in Taiwan. Because this is a long journey, caregivers also require rest in between.

In this study, most participants mentioned that the patients became paralyzed due to accidents. After acute phase treatment and entry into the rehabilitation stage, most family members selected aggressive treatment for patients with emergencies. However, they did not observe improvement in limb function recovery in patients after some time, and then the patients also faced the problem of requiring long-term care. Moreover, their family may have simultaneously experienced changes in the roles between family members. In addition, the interactions between caregivers and patients are sufficient to affect subsequent psychological or physiological improvement in patients. Disease severity is sufficient to affect the life structure of one family, and during the data collection process, we could see that in the face of the disease, the nurses and medical professionals could assist in the physiological care of patients. Furthermore, it is necessary to establish a good doctor–nurse relationship, the support from medical professionals was a good psychological rehabilitation for patients and their family members.

### 5.7. The Power of Rebirth

An arduous journey for patients with SCI experience is from emergency surgery, critical care in the intensive care unit, recuperation in the ward, and rehabilitation in the rehabilitation ward. This is something that people who have never experienced can never understand. In care experiences for patients with SCI, the impact of spinal cord paralysis in patients is particularly serious because it greatly affects daily life as well as work and interpersonal relationships. During interviews, the nurses provided suggestions to patients based on their experiences of caring for patients.

Nurse E mentioned the following:

“*Tips can be shared between patients and family members. The focus of care for everybody is different. Letting patients know that other patients also experience similar things may make them feel better on the inside.*” 

Rehabilitation aims to maximize the remaining capabilities of patients, which requires patients and family members to have an in-depth understanding of the entire rehabilitation process. In addition, homogeneous organizations can jointly build friendly bridges so that patients can walk away from the shadow of the injury.

Similarly, Nurse K also mentioned that:

“*I feel that for clinical recommendations, there are many patient associations in Taiwan. Providing information on these patient associations to family members at suitable times will help diminish the feeling of “why me?” in their minds.*”

People encounter problems in their life, and many of them will have the same thought of “why me?” This may be the reason why it is necessary to establish peer support groups.

In summary, this study explored the experiences of nurses in caring for patients with SCI. Most patients developed paralysis due to different accidents, and most family members chose aggressive treatment for patients’ emergencies. However, when patients are unable to recover to their previous condition, both the patients and their family members will be difficult to accept. Secondly, subsequent rehabilitation is important for limb recovery in patients. This is because limb rehabilitation not only requires a period of 0.5–1 year, but also because rehabilitation can also improve patient confidence. In addition, most nurses are doubtful whether the patient can successfully return to the society, regardless of environmental acceptance, satisfaction, etc. However, few nurses will simultaneously train family members and hemiplegic patients in self-urination and defecation functions in addition to rehabilitation. Decreasing dependence on nurses and accepting guidance from peer support groups can help patients successfully return to work after discharge and manage their finances and life, which will thereby reduce the burden on their families.

## 6. Discussion

In the data analysis, we found that most SCI and paralysis are due to accidents. Limb impairment occurs due to nerve damage. After treatment, patients enter the rehabilitation phase. As the degree of limb function recovery in patients is limited, it is hoped that the needs of the patients and their family members can be met during nursing care. In this section, we discuss the recommendations for paralysis resulting from accidents, acute stage care in paralysis, rehabilitation process in patients after paralysis, changes in family structure, and care experiences.

### 6.1. Dramatic Changes in Life

Based on the 2016 statistics from the Ministry of Health and Welfare, accidents are ranked sixth among the top 10 causes of death in Taiwan. At the same time, accidents are also one of the major causes of SCI [23]. The prevalence of SCI is increasing every year. Around 1200 people develop SCI due to trauma and most patients are young people every year [2,11].

After experiencing traumatic SCI, changes not only occur in physical appearance and cause sensory loss and mobility impairment in patients with SCI; however, it will also affect their physiological, psychological, and social functions [24]. An originally stable psychological state may be greatly affected due to the impact of accidents. In particular, serious permanent injury that is irreversible causes physical disability or even mobility impairment due to paralysis. The transformation of a highly mobile and free person to a person with low mobility and requires assistance from others [18] not only has physical and psychological effects on patients but also causes their family members who are responsible for caring to experience problems such as worry, uncertainty, and role changes [25]. In this study, the nurses also mentioned that paralysis patients experience a tortuous treatment process during the acute phase. At the same time, they must face an uncertain future and mostly require the assistance of others for movement, or even require aids to move their own body. In this study, most of the participants that were cared for by nurses are spinal cord paralysis patients due to trauma. These patients experience major changes in their life. While emergency treatment saved their lives, these patients showed different degrees of recovery, and nurses are required to solve a series of treatment and post-paralysis comorbidity problems.

### 6.2. Life Lost Control

During the acute phase of SCI, the first goal is to prevent permanent SCI and further damage to the spinal cord. Therefore, it is recommended that operations for spinal cord decompression and vertebral fixation be carried out as soon as possible and patients be observed for symptoms of progressive nerve damage [26]. The clinical guideline (spinal cord injury acute management) [27] recommend that orthopaedics, neurosurgical and trauma service should be informed, and neurological assessment and vital sign (including autonomic control) should be performed and monitored intensively. Furthermore, prompt identification of neurogenic shock, before appropriate treatment is carried out. 

Due to damage to the nervous system, acute phase nursing care requires rolling log flipping of the body every 2 h so that the vertebra is straight, for suitable limb activities, and for correct body posture to prevent atrophy in the limb muscles and joint deformities [28]. The content in the aforementioned references is similar to the acute phase care experiences of the interviewed nurses in this study. Patients are admitted to the intensive care unit for observation after surgery and maintenance of limb function is carried out by assisting patients in executing range of motion activities. However, patients with serious nerve damage will experience respiratory training results that do not meet expectations. This causes patients to develop self-harm and depressive behaviors.

Common physiological impairments seen after SCI include hemiplegia and quadriplegia mobility disorders, and abnormalities and loss of physical sensations. In addition, patients tend to develop neurogenic bladder and urinary tract infections. With regards to sexual function, patients may experience reflex erection, erectile dysfunction, and infertility. Abnormalities in the autonomic nervous system will cause patients to be susceptible to orthostatic hypotension, headaches, and loss of physiological functions such as temperature regulation. Therefore, close monitoring of patients is required. In order to alleviate these situations, progressive postural changes can be made and passive range of motion exercises can be carried out to promote reflux blood flow. Hence, nursing professionals must be able to execute correct nervous assessment during the acute phase in patients [27].

However, the study results of Chang et al. (2014) [18] who investigated the prevalence and related factors of post-traumatic stress disorder (PTSD) in patients with SCI found that the prevalence of PTSD in the general population is 5.2% and is 22.4% in patients with SCI. Among these patients with SCI, quadriplegic patients (66.7%) are more prone to PTSD than hemiplegic patients (33.3%). Furthermore, one study explored experiences of cancer patients found that nearly all of the patients accepted that they lived and adapted after being diagnosed with cancer [29]. In this study, the interviewed nurses mentioned that in addition to maintenance of physiological stability when patients are cared in the intensive care unit after acute phase surgery, they face patients who are unable to lift up their limbs and they deal with this situation together with patients. The use of verbal encouragement, range of motion activities, or even observing possible depressive emotions and negative self-concept in patients are common topics in psychological adaptation.

### 6.3. Life after Catastrophic Injury

Zhang (2015) [30] study on experience sharing was conducted with patients with SCI found that medical technology is ever-improving and correct medical intervention can decrease SCI; however, treatment cannot recover patients to the state that they were at before injury. Medical professionals encourage patients to carry out more rehabilitation so that their physical function is maintained to prevent muscle and nerve atrophy. Patients with less severe injury may recover part of their physical function through rehabilitation but this cannot be compared with before injury. In 2011, Babamohamadi, Negarandeh, and Dehghan-Nayeri [31] conducted a study on coping strategies used by people with spinal cord injury. Though qualitative approach, they found three coping strategies of patients in their psychological responses toward paralysis, including religious beliefs (understanding the disease as a divine fate and as a spiritual combat), hope, and making efforts toward independence/self-care.

Peter et al. (2012) [4] indicated that strengthening the patient’s psychology is the initial goal in patient recovery in clinical practice. Employing supportive strength so that patients are able to adjust their psychology when faced with disability. Furthermore, it will allow them to gradually approach their best state and encourage patients to step out to society successfully. A study found that patients with SCI who were injured for more than two years, they have received occupational therapy for SCI can adapt better to future work. It is recommended that patients should be encouraged to contact the SCI society and related groups as soon as possible for life care. Further, occupational therapy should be also utilized as this will aid in psychological and social adaptation [32].

Most SCI due to accidents were found in this study. Therefore, patients must accept the fact that they will be wheelchair-bound or bedridden in the future and the severity of the injury will cause them to be unable to physically and psychologically adapt in a short amount of time or even return to work. Some hemiplegic patients can successfully find employment with the assistance of their family members. In this study, some patients also successfully returned to society after counseling. However, some patients are unwilling to reenter society due to society’s view of them and their requirements for equality in work.

In addition to changes in the relationship between family members, individuals also face extreme stress and challenges. One study evaluating sexual function education for patients after SCI found that patients would like the chance to discuss this topic during their rehabilitation process [32]. The information provided by nurses on sex to patients is extremely limited, but sexual activities are also a part of recovery for patients. Insufficient sex counseling and education may somewhat affect the patient’s quality of life [33]. A survey on the sexual needs and coping strategies in patients with SCI was conducted in Taiwan in 2016. Results showed that for male SCI patients, more value should be put on sexual needs for SCI patients. Male patients are more proactive in their coping strategies toward physiological needs in sexuality. In contrast, women are more negative. This is because in Eastern society, it is normal for men to have sexual needs but shameful for women to have this requirement. This causes women to be repressed and feel helpless and wronged when they have needs in sexuality. In this study, the participants mentioned the physiological needs in sexuality after suffering from SCI, but they did not mention the sexual satisfaction of patients. At the same time, this also causes the researchers to recognize that patients are also humans, sex in the entire course of SCI is also a topic that needs to be explored. On the other hand, when nurses encounter patients’ sexual dysfunction in SCI, they should have sharp observation skills and correct concepts and countermeasures so that nurse professionals can carry out its roles to their best.

### 6.4. Life Takes Turns for the Worse for Family

The Republic of China SCI Patient Alliance (2016) [34] pointed out that Taiwanese families’ value morals, and believe that caring for family members is natural and right. Therefore, it is often neglected that the fact that family caregivers also require assistance and support. Most of the time, the role of the family caregiver will fall on the most disadvantaged person in the family and society. Our study found that after paralysis, most patients will develop psychological problems due to their disease condition. Further, the recovery of patients can be affected. In addition to staying at the intensive care unit for continuous observation, nurses must simultaneously face the emotions of the patients when they first learn of their paralysis and provide encouragement for patients.

A scoping review (including both quantitative and qualitative studies) found that caregivers’ burnout is one of the barriers in taking care of patients following SCI [34]. Some studies have indicated that half of the family members felt that they are unable to cope with subsequent care of patients. Therefore, nurses should proactively discover such problems at the acute stage and provide disease-related information to patients and their family members. Moreover, establishing good nurse–patient relationships among the medical team, patients, and family members, it is able to achieve reorganization of patients and their families [35,36]. In addition, reflection on the meaning of life is also a family member’s goal during taking care of their love one who has SCI [37]. Furthermore, one study focusing on adolescents with SCI looked at the effect of family-centered education on the quality of life with spinal cord injuries, and found that it was essential to have necessary grounds for training families for adolescents with SCI by nurses to improve their quality of life and conduct research on their problems [38].

In addition, the grounded theory study of Pullin and Mc Kenzie (2017) [39] explored the life experiences of long-term caregivers of spinal cord injuries. They found that there is a need for substitute for the role of the primary caregiver in the family so that they can take a temporary rest. They hope that society can provide a caring service to address the loneliness, isolation, and grief in caregivers. Chuang (2010) [40] studied social support, leisure adjustment, and burden in primary caregivers of patients with SCI. They found that social support and leisure adjustment will help to decrease caregiver burden. This may be due to psychological, physiological, and identity differences when caring for patients with SCI. Different needs and caregiver burden can be alleviated by meeting the needs of the caregivers or taking turns to care for patients. This study found that caring for one SCI patient poses multiple challenges, of which the physical and mental health of the primary caregiver shows an evident relative relationship. In this study, nurses assisted patients in limb training and provided psychological support and encouragement during treatment.

### 6.5. The Power of Rebirth

The study of Lee, Cheng, and Chiou (2014) [41] explored the employment process of patients with SCI and recommended that patients with SCI should actively participate in related supportive social activities to understand their physiological limitations, prognosis, and undergo psychological and social adjustments through sharing with other patients or health professionals. Furthermore, the power of rebirth in this study is similar to cancer patients’ post-traumatic growth, which opens new possibilities to their life [42]. In addition, Lee et al. (2017) [16] indicated that patients must be trained on independent life functions as early as possible during the early stage of inpatient rehabilitation, and patient societies can be used or discharge preparation services can be strengthened. The sharing of experiences with fellow patients with SCI will give hope to patients. It is recommended that group experience sharing can be regularly held or organized for patients with SCI, or assistance can be rendered for referral to SCI patient groups [43]. In addition to providing physical and mental, social, spiritual care, and rehabilitation treatment for patients in clinical care, the post-discharge care of patients can be further extended to the community so that care becomes continuous holistic care [44]. This study found that there are many similarities when the aforementioned care recommendations are compared with the care experiences of patients with SCI by nurses. The perspective of empathy is used to think about caring for patients, communication, and accompaniment of family members. It is expected that patient groups and the SCI groups can be used to provide learning and encouragement environments. Companionship and support are needed when patients depend on themselves to overcome disability and find the life that originally belonged to them. This study provides some insights we have known from prior research. In addition, this study further explores the experiences of caring for patients with spinal cord injury through Taiwan nurses’ perspective. It makes a contribution to the field and the nursing literature.

## 7. Strengths and Limitations

From this study, we found that five themes, which are the five different lives of patients after suffering from SCI: “dramatic changes in life”, “life lost control”, “life after injury”, “life takes turns for the worse for family”, and “the power of rebirth”. Nurses accept that guidance from peer support groups can help patients successfully return to work after discharge, and manage their finances and life. In addition, nurse’s experiences can be used to assist other clinical nurses to understand the clinical experience when taking care of patients with SCI. Furthermore, nurses can focus on different aims in taking care of SCI patients at different stages, such as physiological function at the acute stage, preventing limb spasm at the rehabilitation stage, and employment counseling to improve the confidence of patients in returning to society.

The limitations of this study were that the study participants were nurses from one regional teaching hospital and their ages ranged from 23 to 40 years old, which is quite young. In addition, only one male nurse was included in this study. Therefore, the results of this study cannot completely explain the experiences of caring for patients with SCI by nurses aged ≥ 40 years or male nurses. During the study, we found that most patients experienced SCI due to accidents, and spiritual and social care is easily overlooked in patients. It is recommended that case managers be set up to follow-up on the subsequent care of patients, peer support groups be employed to increase human contact for patients, and an environment can be set up for work counseling.

## 8. Conclusions

This study helps clinical nurses to understand the overall phenomenon of caring for patients with SCI. Their personal experiences are used to assist clinical nurses to understand the clinical experience when there is mental adaptation in patients with SCI. Furthermore, it can be used as a reference for future care and clinical education to improve care quality toward SCI patients. Our study found that in the experience of caring for patients with SCI in nurses, there are some minor differences in care at different stages. An example is the focus on resuscitation and maintenance of physiological function at the acute stage, in comparison to the focus on maintenance of remaining limb function and preventing limb spasm at the rehabilitation stage. In addition, employment and association intervention and counseling for patients can improve the confidence of patients in returning to society.

In clinical practice recommendations, dedicated case managers can follow-up on subsequent care in patients, and assistance can be provided to establish patient support groups in the hospital so that patients and family members can mutually support each other. In addition, on service training for psychological and sex education for patients can be increased in clinical nurses. This will not only improve the care quality in nursing education but also achieve holistic care. This study suggested that future studies can explore the adaptation process of patients after SCI and also the coping struggles in family members. In addition, the sexual need of patients with SCI is another topic to be explored. It is hoped that health care professionals can further understand the overall experience of patients with SCI.

## Figures and Tables

**Table 1 ijerph-19-01058-t001:** Demographic information of the participants.

S/N	Sex	Age	Marital Status	Religion	Education Level	Number of Years of Clinical Nursing Experience	Number of Years of Experience in Caring for Spinal Cord Injuries	Current Clinical Unit
A	Female	27	Single	Taoist	University	4	4	Neurosurgery intensive care unit
B	Female	34	Married	Buddhist	University	14	9	Organ coordinator (neurosurgery intensive care unit, general surgery intensive care unit)
C	Female	33	Married	Taoist	University	10	10	Hospice case manager (neurosurgery intensive care unit)
D	Female	34	Married	Taoist	University	14	13	Internal medicine ward
E	Female	26	Single	Taoist	University	2	2	Neurosurgery intensive care unit
F	Female	36	Married	Taoist	University	16	16	Rehabilitation ward
G	Female	39	Married	Free thinker	University	16	16	Rehabilitation ward
H	Female	38	Married	Free thinker	Junior college	15	15	Rehabilitation ward
I	Female	40	Married	Free thinker	University	20	20	Rehabilitation ward (Nurse practitioner)
J	Female	38	Married	Taoist	University	14	14	Rehabilitation ward
K	Female	40	Married	Free thinker	Junior college	18	18	Case manager (neurosurgery intensive care unit, respiratory care center)
L	Male	23	Single	Christianity	University	3	3	Neurosurgery ward
M	Female	33	Married	Free thinker	University	11	11	Neurosurgery ward
N	Female	35	Married	Free thinker	University	15	15	Chronic respiratory care ward
O	Female	30	Married	Free thinker	University	10	10	Case manager (was previously in neurosurgery intensive care unit)

## Data Availability

Data supporting reported results can be found by request, please contact Shou-Yu Wang (email: cindy.wang@une.edu.au) or Shih-Ru Hong (email: sea198083@yahoo.com.tw).

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
