# Peer review of "Five Different Lives after Suffering from Spinal Cord Injury: The Experiences of Nurses Who Take Care of Spinal Cord Injury Patients"

_ijerph, 2022, doi:10.3390/ijerph19031058_

Round 1
Reviewer 1 Report
I would like to thank you for submitting and give me the opportunity to review the manuscript entitled: " Five different lives after suffering from spinal cord injury: The experiences of nurses who take care of spinal cord injury patients". The research topic undertaken by the authors is interesting and may be of great importance for increasing the nurses' knowledge about how to deal with the comprehensive care of patients with spinal cord injury (SCI) and to know aspects that should not be neglected. This study should be of interested for the journal readers. However, at the moment the article needs significant revisions in order for its acceptance in IJERPH to be considered.
Major concerns:
The number of participants is extremely low, (especially in the group of men) for a primarily qualitative study regarding nursing experience in the care of SCI patients. Even knowing the predominance of the female gender in the nursing discipline, the lack of parity in this work must be solved. The experience and the way of facing the same event by both genders can be extremely different. Therefore, it is not enough to list this as a study limitation.
In addition, and although the rigor of this study is corroborated and it is based on the elements proposed by Lincoln & Guba, it is necessary and would greatly enhance the study to show some quantitative data on the responses and experiences of nurses in the five topics studied “dramatic changes in life”, “life lost control”, “life after injury”, “life takes turns for the worse for family”, and “the power of rebirth”. In each section only, the experience of some nurses is transcribed. By way of summary, an overall quantitative analysis of the experiences revealed by the study participants is strongly recommended for the review.
Minor concerns:
Although it is mentioned in the limitations of the study, it would be necessary to incorporate in the text the importance of age and nursing experience with respect to the validity of the answers and experiences mentioned. Similarly, the nursing experience will be different according to the patient's level of dysfunctionality, so I consider it essential to mention this variable in the text and in the interpretation of the results.
Since this is a mainly qualitative opinion study, it is necessary to detail whether the same researchers conduct the interviews and analyze the data. It is important not to commit bias in the interpretation of data. On the other hand, how do you ensure the interviewer's objectivity during the interviews?
In the results section, table 1 and its explanation in the text of the article is redundant. I recommend deleting one of the two.
It is recommended that the bibliographical references be revised. There are some formatting errors. Likewise, the doi is missing in many references and some words are in bold.
Finally, some unimportant errors, such as the lack of a dot on line 33 after [6], larger font size on line 56 and the text on line 248 is in bold type.
Author Response
Response to reviewers’ comments for ijerph 1505425
Reviewer #1:
I would like to thank you for submitting and give me the opportunity to review the manuscript entitled: " Five different lives after suffering from spinal cord injury: The experiences of nurses who take care of spinal cord injury patients". The research topic undertaken by the authors is interesting and may be of great importance for increasing the nurses' knowledge about how to deal with the comprehensive care of patients with spinal cord injury (SCI) and to know aspects that should not be neglected. This study should be of interested for the journal readers.
Response: Thanks for the reviewer’s comments.
However, at the moment the article needs significant revisions in order for its acceptance in IJERPH to be considered.
Major concerns:
The number of participants is extremely low, (especially in the group of men) for a primarily qualitative study regarding nursing experience in the care of SCI patients. Even knowing the predominance of the female gender in the nursing discipline, the lack of parity in this work must be solved. The experience and the way of facing the same event by both genders can be extremely different. Therefore, it is not enough to list this as a study limitation.
Response: Thanks for the reviewer’s comments. During the data collection phase, we did attempt to recruit male nurses. However, some male nurses either cancel their interviews or are not able to do the interviews due to military service. Therefore, we list it in the study limitation. We had added this statement in P. 4-5, line 168-170.
In addition, and although the rigor of this study is corroborated and it is based on the elements proposed by Lincoln & Guba, it is necessary and would greatly enhance the study to show some quantitative data on the responses and experiences of nurses in the five topics studied “dramatic changes in life”, “life lost control”, “life after injury”, “life takes turns for the worse for family”, and “the power of rebirth”. In each section only, the experience of some nurses is transcribed. By way of summary, an overall quantitative analysis of the experiences revealed by the study participants is strongly recommended for the review.
Response: We did find some quantitative data on the responses and experiences of this study, but not many. We used other qualitative studies to reinforce our themes. In addition, the themes that we found in this study can be seen as a whole experience and interconnect with each other represent nurses’ experiences in taking care of SCI patients. We had added new content in the discussion section. Thanks for the reviewer’s recommendations. Please see P. 10, line 422-424, P. 11, line 490-491.
Minor concerns:
Although it is mentioned in the limitations of the study, it would be necessary to incorporate in the text the importance of age and nursing experience with respect to the validity of the answers and experiences mentioned. Similarly, the nursing experience will be different according to the patient's level of dysfunctionality, so I consider it essential to mention this variable in the text and in the interpretation of the results.
Response: In this study, we put “Currently employed nurses with ≥6 months of clinical experience” in the inclusion criteria. The participants’ age is between 27-40 years old and years of clinical nursing experience is between 2-20 years. The patient's level of dysfunctionality in nurses who participated in this study is between half paralysis to the whole paralysis. However, we see participants’ interpretation of their experiences in taking care of SCI patients as a whole social process. Therefore, we do not analyse the data according to their age, experiences, and the patient's level of dysfunctionality. We put them into limitations. Thanks for the reviewer’s recommendations.
Since this is a mainly qualitative opinion study, it is necessary to detail whether the same researchers conduct the interviews and analyze the data. It is important not to commit bias in the interpretation of data. On the other hand, how do you ensure the interviewer's objectivity during the interviews?
Response: All interviews were done by the second author in this study. All authors participated in the data analysis process. The authors met regularly to discuss data analysis to avoid bias in the interpretation of data. By meeting regularly during the data analysis phase (data collection and data analysis happened concurrently in the study process), it is able to ensure to minimise the interviewer's objectivity during the interviews. Thanks for the reviewer’s recommendations. Please see P. 3, line 107-112.
In the results section, table 1 and its explanation in the text of the article is redundant. I recommend deleting one of the two.
Response: Thanks for the reviewer’s recommendations. We had deleted table 1 and change table 2 to table 1.
It is recommended that the bibliographical references be revised. There are some formatting errors. Likewise, the doi is missing in many references and some words are in bold. Finally, some unimportant errors, such as the lack of a dot on line 33 after [6], larger font size on line 56 and the text on line 248 is in bold type.
Response: We had added doi to the reference section. However, doi cannot be found in some references. We also amended some minors. Thanks for the reviewer’s recommendations. Please see PP. 13-15.

Reviewer 2 Report
The Authors investigated open issues, difficulties and specific needs of clinical nurses involved in the healthcare of patients experiencing spianl injuries. The authors employed a qualitative research approach. The article is interestign and timely. I have comments:
- Please state if you reached data saturation. In this case, please describe the methods by which you ensured data saturation to be reached.
- The power of rebirth. I suggest to introduce and discuss the concept of post-traumatic growth, which has been described in different medical conditions, such a s cancer as an example.
Author Response
Response to reviewers’ comments for ijerph 1505425
Reviewer #2:
The Authors investigated open issues, difficulties, and specific needs of clinical nurses involved in the healthcare of patients experiencing spinal injuries. The authors employed a qualitative research approach.
Response: Thanks for the reviewer’s comments.
Please state if you reached data saturation. In this case, please describe the methods by which you ensured data saturation to be reached.
Response: Thanks for the reviewer’s comments. This study did reach data saturation. During the data analysis process, the themes that we found are repeated shown in the data analysis. Please see P. 3, line 133-135.
The power of rebirth. I suggest introducing and discussing the concept of post-traumatic growth, which has been described in different medical conditions, such as cancer as an example.
Response: Thanks for the reviewer’s comments. We had added content in the discussion section regarding post-traumatic growth in cancer conditions and compare it with this study. Please see P. 12, line 552-553.

Round 2
Reviewer 1 Report
The authors have improved the manuscript and justified the reviewer's questions so I suggest accepting the manuscript. Anyway I suggest minimal typographical changes that have not been corrected as, in line 174 should change table 2 to table 1, since table 2 has been deleted. In line 491 there are two "in" and in line 38 the period after the reference [6] is still missing.
Author Response
Reviewer #1:
The authors have improved the manuscript and justified the reviewer's questions, so I suggest accepting the manuscript.
Response: Thanks for the reviewer’s comments.
Anyway, I suggest minimal typographical changes that have not been corrected as, in line 174 should change table 2 to table 1, since table 2 has been deleted. In line 491 there are two "in" and in line 38 the period after the reference [6] is still missing.
Response: We had amended these minimal typographical errors (marked in green). Please see P. 2, 4, 11. Thanks for the reviewer’s comments.

Reviewer 2 Report
I do not have any further comment.
Author Response
Reviewer #2:
I do not have any further comments.
Response: Thanks for the reviewer’s comments.
